# Mutational Landscape and Actionable Target Rates on Advanced Stage Refractory Cancer Patients: A Multicenter Chilean Experience

**DOI:** 10.3390/jpm12020195

**Published:** 2022-01-31

**Authors:** Miguel Cordova-Delgado, Mauricio P. Pinto, Carlos Regonesi, Luis Cereceda, José Miguel Reyes, Laura Itriago, Alejandro Majlis, Pablo Rodríguez, André Fassler, Mauricio Mahave, María Elisa León, Jorge Gallardo, María Paz Rodríguez Z., Alejandro Berkovits, Patricio Manque, Juvenal A. Ríos, Benjamín Garcia-Bloj, Marcelo Garrido

**Affiliations:** 1Department of Hematology and Oncology, Faculty of Medicine, Pontificia Universidad Católica de Chile, Santiago 8330032, Chile; cordova.delgado.m@gmail.com (M.C.-D.); mauricio_pinto@outlook.com (M.P.P.); 2Faculty of Chemical and Pharmaceutical Sciences, Universidad de Chile, Santiago 8380494, Chile; 3Division of Medical Oncology, Clínica Las Condes, Santiago 7591047, Chile; cregonesim@clinicalascondes.cl (C.R.); lcereceda@clc.cl (L.C.); jmreyes@clinicalascondes.cl (J.M.R.); litriago@clinicalascondes.cl (L.I.); amajlis@clinicalascondes.cl (A.M.); 4Oncology Department, Clínica IRAM, GESMED, Santiago 7630370, Chile; pablormonarca@yahoo.com; 5Clínica Dávila, Santiago 8420384, Chile; dr.fassler@gmail.com; 6Fundación Arturo Lopez Perez, Santiago 7500855, Chile; mahavem@falp.org; 7Medical Oncology, Clínica Reñaca, Viña Del Mar 2540364, Chile; mleonpr@gmail.com; 8Clínica Indisa, Santiago 7520440, Chile; dr.jgallard@gmail.com; 9Centro de Oncologia de Precision, Escuela de Medicina, Universidad mayor, Santiago 7560908, Chile; mpaz.rodz@gmail.com (M.P.R.Z.); aberkovits@inmunocel.cl (A.B.); patricio.manque@umayor.cl (P.M.); juvenal.rios@umayor.cl (J.A.R.); benjamin.garcia@umayor.cl (B.G.-B.)

**Keywords:** precision oncology, Next-Generation Sequencing, on-label, off-label, therapy

## Abstract

Major advances in sequencing technologies and targeted therapies have accelerated the incorporation of oncology into the era of precision medicine and “biomarker-driven” treatments. However, the impact of this approach on the everyday clinic has yet to be determined. Most precision oncology reports are based on developed countries and usually involve metastatic, hard-to-treat or incurable cancer patients. Moreover, in many cases race and ethnicity in these studies is commonly unreported and real-world evidence in this topic is scarce. Herein, we report data from a total of 202 Chilean advanced stage refractory cancer patients. Retrospectively, we collected patient data from NGS tests and IHC in order to determine the proportion of patients that would benefit from targeted treatments. Overall >20 tumor types were included in our cohort and 37% of patients (*n* = 74) displayed potentially actionable alterations, including on-label, off-label and immune checkpoint inhibitor recommendations. Our findings were in-line with previous reports such as the cancer genome atlas (TCGA). To our knowledge, this is the first report of its kind in Latin America delivering real-world evidence to estimate the percentage of refractory tumor patients that might benefit from precision oncology. Although this approach is still in its infancy in Chile, we strongly encourage the implementation of mutational tumor boards in our country in order to provide more therapeutic options for advanced stage refractory patients.

## 1. Introduction

The ultimate goal of precision oncology is to deliver the right cancer treatment to the right patient in a timely manner. In recent years, the rapid development of massively parallel sequencing technologies (also called next generation sequencing; NGS) and the progressive reduction of its associated costs have allowed characterization of the mutational landscape of cancer genomes, identifying “actionable” targets and “driver” mutations responsible for tumor growth and progression. Furthermore, the combined use of NGS gene-panels that expand the number of genes that can be simultaneously analyzed, along with immunohistochemistry (IHC)-based techniques, increases the possibility of finding a matching targeted therapy. Successful examples include anti-HER2 therapy for HER2/neu overexpressing gastric or breast cancer patients [1,2,3,4,5,6], the use of vemurafenib for BRAF-V600E mutant metastatic melanoma patients [7,8,9], or PARP-inhibitors for BRCA-mutant advanced ovarian [10], metastatic breast [11,12] and metastatic pancreatic adenocarcinoma patients [13,14].

Despite major advances in this field, the impact and clinical utility of this approach especially on advanced stage patient survival remains sparse. In most cases, phase II clinical trials demonstrate the effectiveness of targeted therapies. Unfortunately, these studies are commonly focused on a small subset of cancer patients that harbor specific alterations and/or clinical characteristics. In contrast, precision oncology reports on real world data are somewhat scarce. These reports are mainly based on North American, European and Asian countries. Furthermore, race is often unreported in clinical–genomic studies; thereby minorities are usually underrepresented. In fact, NGS studies based in Latin America are extremely scant. Given the geographical heterogeneity observed for some malignancies, precision-oncology real-world evidence studies in specific regions could imply a differential impact of this approach on these geographical areas.

Herein, we retrospectively analyzed the clinical utility of tumor profiling by NGS/IHC in the routine practice across several Chilean tertiary health centers. We also assessed the final decision made by oncologists following the recommendation. To our knowledge, this is the first report of its kind in Latin America.

## 2. Materials and Methods

### 2.1. Study Design, Patients and Tumor Molecular Profile

This research was approved by the scientific ethics committee affiliated to the school of medicine of the Pontificia Universidad Catolica de Chile (approval ID#200121002, dated on 30 April 2020). Patients diagnosed with advanced or metastatic solid cancer and who underwent tumor genomic profiling were eligible for inclusion in this study. A total of 202 patients were included. The different platforms used for tumor characterization are included in Appendix A. Briefly, in 202 patients the tumor DNA was sequenced through NGS (different number of genes depending on the platform) and, of these, in 117 the result also included the tumor mutational burden (TMB). Furthermore, in 124 patients microsatellite status was determined using DNA sequencing (NGS or Sanger). Finally, PDL1 and HER2 expression was determined using immunohistochemistry (IHC) in 126 and 128 patients, respectively. Commercial companies independently performed all assays (Appendix A). Details of each of the platforms used can be found in Appendix A. Information on cancer type, platform, and type of sample used is also included in Appendix A.

### 2.2. Criteria for Actionability

In order to establish the potential clinical applicability of the genomic alteration found in each patient, we used OncoKB [15], which is an expert-guided precision oncology knowledge database that assigns each variant to different levels of evidence corresponding to its actionability. We selected high levels (1 and 2) of evidence to establish the potential recommendations on label and off label, where level 1 corresponds to a FDA-recognized biomarker predictive of response to an FDA-approved drug in this indication and level 2 corresponds to a standard of care biomarker recommended by the NCCN or other expert panels, predictive of response to an FDA-approved drug in this indication [15]. Furthermore, we established whether patients could potentially benefit from immunotherapy, according to the following criteria: (1) microsatellite instability (MSI), (2) High/Medium expression of PDL1 according to molecular report and (3) High/Medium TMB, TMB-level defined as low ≤ 5 mut/Mb), intermediate > 5 mut/Mb, ≤ 15 mut/Mb), or high > 15 mut/Mb, based on the thresholds described previously [16].

### 2.3. Comparison of Molecular Alterations to the Cancer Genome Atlas (TCGA) Cohorts

A descriptive comparison of the frequency of molecular alterations between the six main tumor types in our study and matched cohorts of the TCGA was performed. The frequencies were obtained using the bio-portal database [17]. The studies selected for each type of tumor were: colorectal cancer [18], pancreatic cancer [19], sarcoma [20], breast cancer [21,22], ovarian cancer [23], lung cancer [24].

### 2.4. Oncologists’ Survey

The usefulness of the molecular test in clinical decision was evaluated in a subset of 46 patients through a survey applied to oncologists. The question was: “Has the treatment decision changed based on the results generated by the molecular test?”.

## 3. Results

A total of 202 advanced stage patients were included in our study. Main cancer types were gastrointestinal (*n* = 92; 45%) followed by sarcomas/unknown primary (*n* = 36; 18%), and gynecological/breast (*n* = 33; 16%). The detailed distribution of tumor types is summarized in Table 1. Overall, 77% of patients had at least one alteration confirmed by NGS.

First, we sought to determine the mutational landscape of tumors. Our workflow is depicted in Appendix A. Briefly, the entire cohort (*n* = 202) was analyzed by NGS using gene panels at various platforms. Patient subsets were analyzed by IHC to determine HER2 (*n* = 126) and PDL1 (*n* = 128) expression. MSI status and TMB were also analyzed by a combination of NGS/IHC or NGS in patient subsets (*n* = 124 and *n* = 117, respectively). Mutational landscapes on analyzed tumors are displayed in Figure 1. As pointed out, gastrointestinal cancers were the most prevalent in our cohort. Among these, *KRAS* (51%) was the most frequently altered gene, mainly due to colorectal and pancreatic cancers followed by *TP53*, *APC*, *ARID1A*, *SMAD4* and *PIK3CA*. Please note the low number of alterations in gallbladder and hepatic tumors. For genitourinary tumors *ERBB2* and *KIT* were the most commonly altered genes, but the number of alterations in these tumors was generally low. As expected, *TP53* was the most frequently altered gene on gynecological tumors, followed by *ARID1A* and *PIK3CA*. Similarly, *BRCA1* and *BRCA2* truncations were found in these tumors. As expected *PIK3CA* mutations and *CCND1* amplifications were also recurrent among breast cancer patients; *TP53* was frequently altered in lung cancer patients and in those with unknown primary tumors. On the other hand, sarcomas were characterized by a low number of alterations. Other less prevalent tumor types analyzed included CNS and NETs.

Next, we sought to compare these results with the TCGA database. Figure 2 shows that the frequency of most gene alterations was similar. However, there were some notorious exceptions on well-known relevant genes for certain cancer types, for example, *KRAS* and *APC* in colorectal cancer: 55% and 50% in our cohort vs. 35% and 75% in TCGA, respectively; also, *BRCA2* and *SMAD4* in pancreatic cancer: 8% and 8% in our cohort vs. 2% and 23% in TCGA; *EPHA2* on sarcoma: 11% in our cohort vs. 0%; *TP53* and *ESR1* in breast cancer: 13% and 13% in our cohort vs. 34% and 0% in TCGA. Other examples were *TP53* and *ARID1A* in ovarian cancer, and *TP53* and *KRAS* in lung cancer.

Our tumor analyses also included immunohistochemistry for certain factors (Appendix A). In particular lung and pancreatic cancers displayed medium/high levels of PDL1 expression. Interestingly, PDL1 positivity correlated with HER2+ in bladder cancer patients. Additionally, we determined TMB and microsatellite instability in a subset of patients; in general colorectal cancer patients showed the highest positivity for these markers (Figure 1).

Based on this molecular analysis, 75 out of 202 (37%) patients received informed treatment recommendations that included on-label, off-label and immune checkpoint inhibitor (ICI) recommendations. Overall, 18 out of 202 treatment recommendations (9%) were on-label therapies; within this group, the fulvestrant + alpelisib combination and PARP inhibitors were the most frequently recommended with 39% and 17%, respectively (Figure 3A, left panel). On the other hand, 38 patients (19%) received off-label recommendations. Among these, anti-PI3K, PARP inhibitors and imatinib were the most commonly recommended with 32%, 21% and 18%, respectively (Figure 3A, right panel). Then, ICIs were recommended in 42 out of 202 patients (21%), based on medium/high PDL1 expression, medium/high TMB or MSI status (Figure 3B). Next, we analyzed the percentage of recommendations by cancer type. As shown in Figure 3C the highest percentage of on-label recommendation was observed in breast, followed by ovarian cancer patients. These tumors also displayed the highest rates of overall recommendations (on-label, off-label and ICI). Similarly, lung and colorectal cancer patients had relatively high levels of recommendation. In contrast, pancreatic tumors and sarcomas had the lowest rates of therapy recommendation based on actionable alterations.

Next, we analyzed a small subset of patients (*n* = 35) from our cohort who had available clinical information. Among these, only four received targeted therapies based on molecular testing: one of them displayed stable disease (to Nivolumab-Ipilimumab), one had a partial response to Olaparib, one had a complete response (to Imatinib) and the other progressed after Regorafenib. The characteristics and outcomes in these patients are summarized in Appendix A. On the other hand, the clinical characteristics, previous lines of therapy and time between the diagnosis and the molecular test of the patients who did not receive target therapy are detailed in Appendix A.

Finally, we applied a brief survey to a group of oncologists that made treatment decisions in a subset of 46 patients. The survey asked about the influence of molecular profiling upon treatment choices. Overall, oncologists estimated that molecular testing was useful for clinical management in 50% of the evaluated cases.

## 4. Discussion

Major advances in massively parallel high-throughput sequencing technologies and the concomitant development of molecularly targeted therapies have shifted the paradigm in oncology from “one size fits all-standard of care” to “biomarker-driven” treatments. Despite this, the true impact of precision oncology in everyday clinical practice remains to be determined. Previous studies have sought to quantify the contribution of these strategies, but several factors must be taken into consideration for such analysis. A German single-center retrospective study analyzed a total of 198 metastatic cancer patients in the context of implementation of a molecular tumor board [25]. Specific treatment recommendations were given to a subset of 104 patients, which included 9.6% off-label targeted therapies (*n* = 19), 9% combined treatments (*n* = 18), 18.1% ICI (*n* = 36) and 6.5% trial inclusions (*n* = 13). Importantly, in 68% of these cases (*n* = 71) recommendations were not implemented. A second, similar study retrospectively analyzed 600 incurable solid-tumor patients; 51.7% (*n* = 310) received therapy recommendations and 15.8% (*n* = 95) received genomic-driven recommendations; from these 7.09% (*n* = 22) were on-label and 14.9% (*n* = 45) were off-label [26]. Surprisingly, and despite the heterogeneity of our cohort, these percentages are in line with our findings which indicated 9% of on-label, 19% off-label and 21% of ICI therapy recommendations (Figure 3). In terms of actionable targets, our cohort displayed a 37% rate of actionability; this is also in line with a recent report of a genomically guided clinical trial coordinated by the NCI that collected a total of 5954 refractory tumor biopsies across >1000 centers [27]. This trial found actionable targets in 37.6% of patients and tested the feasibility of large-scale screenings of molecular alterations by NGS.

Clearly, the approach of our study, which combined NGS and IHC, increases actionability rates in our cohort. Other studies confirm that the addition of IHC or other tests increases the usefulness of the information obtained from NGS studies alone [28]. As noted, a survey applied to oncologists revealed that 50% estimated that molecular testing was useful for clinical management. Similarly, the above mentioned study indicates that 60.4% of the oncologists followed the recommendations of molecular profile reports. Furthermore, a nationally representative survey of oncologists in the US found that 75.6% use NGS tests to guide treatment decisions, including those for advanced refractory cancers, for clinical trial eligibility and for off-label use of therapeutic agents [29].

As explained above, precision oncology reports are usually based on developed countries. Moreover, patient race in these studies goes usually unreported. A systematic review that included a total of >15,000 patients across 231 studies found that only 37% and 17% reported race and ethnicity, respectively [30]. Within this context, Latin American-based reports are extremely rare. To our knowledge, our report is the first of its kind within the region delivering real-world evidence on the mutational landscape in a cohort of 202 Latin American-ancestry advanced stage cancer patients. Our data included >20 tumor types, and besides NGS tests we assessed HER2 and PDL1 expression, MSI status and TMB in specific subsets. As shown in Figure 2, when we analyzed the most frequently altered genes by cancer type in our cohort versus TCGA, we found some coincidences but also some notorious discrepancies. These can be attributed to some of the limitations of our study (listed below) including number of samples, differences in NGS panels (as shown in Appendix A and Appendix A), tumor stage and previous treatments received by the patients. As an example, the high frequency of *ESR1* in our cohort versus TCGA is probably due to the inclusion of a patient with luminal tumors (ER+) who received hormone therapy, and therefore represents a selection bias. Previously, other authors have analyzed the main obstacles to the implementation of precision medicine programs in the region; these are mainly related to high costs and limited (or lack of) access to health coverage [31]. These studies also emphasize the need for locally generated data. Evidently, ethnic and lifestyle influences translate into a wide spectrum of somatic mutations and gene polymorphisms causing differences in biomarkers and treatment response. In this regard, mutational tumor boards inside health institutions are yet to be implemented in Chile and should be encouraged, especially for high incidence/mortality malignancies such as gastric cancers.

Perhaps the major limitation of our study is the lack of data on therapeutic impact. However, we obtained clinical information for a small subset of patients (Appendix A). Interestingly, the average number of therapies received in this subset was 3.1 and the average elapsed time between diagnosis and molecular testing was 34.9 months. This might explain the low number of patients that are effectively treated with targeted therapies. Furthermore, these patients usually have a very limited set of therapeutic alternatives after several lines of treatment. This also explains the negative results, or the lack of benefit reported in several studies. Throughout the literature the debate on this topic is still ongoing; in 2015 a large French open-label phase II trial sought to determine the impact of molecularly targeted therapies on patient survival. This study involved eight centers and enrolled a total of 741 metastatic refractory cancers (any type). From these, 195 patients were randomly assigned to either control (*n* = 96) or experimental (*n* = 99) arms [32]. Investigators found no benefit in terms of PFS by the use of molecularly targeted therapies versus the physician’s choice. Furthermore, the study recommended enrollment into clinical trials rather than the off-label use of these agents. Importantly, targeted therapies in this study were restricted to 10 available regimens. Subsequently, in 2017 a single-center single-arm trial in the US enrolled a total of 1035 adult advanced hard-to treat cancer patients. Based on genomic analyses a subset of 199 patients were scheduled to received matched therapy according to actionable targets [33]. The clinical benefit was measured as a ratio of PFS on matched therapy (called PFS2) over PFS on prior therapy (called PFS1), setting a threshold at PFS2/PFS1 = 1.3. Overall, 193 patients were evaluated for PFS and 33% (*n* = 63) had a PFS2/PFS1 > 1.3. Obviously, these results should be further validated, but they suggest that at least a fraction of hard-to-treat patients could benefit from this approach. More recently, a smaller single-center study compared the outcomes of a group of 22 metastatic cancer patients which received genomically-guided targeted therapies versus 22 matched control patients treated with either chemotherapy or best supportive care [34]. This study demonstrated a significant increase in median OS by the use of targeted therapies. Interestingly, the study also found that patients under targeted therapies had lower weekly costs during their treatment versus controls. Other limitations of our work include the lack of patients’ clinical data such as tumor histology or patient performance status. Furthermore, (1) This was a highly heterogeneous convenience sample, and in many cases the numbers are too small. (2) Data were collected from private health centers. In general, these centers serve higher income patients. On the other hand the Chilean population is highly heterogeneous with a strong European ancestry. Unfortunately this information could not be collected for this study and therefore could represent another bias. Evidently, the frequencies and/or the distribution of cancer types may not reflect the Chilean reality. (3) Finally, as shown by Appendix A, a variety of NGS assays were used for this cohort, which further increases the heterogeneity of the sample and may also represent an analysis bias.

## 5. Conclusions

In summary, our study provides the first Latin American-based precision oncology report delivering real-world evidence. Our data suggest that a 37% of advanced refractory cancer patients displayed actionable targets. Therefore, as suggested by others, our findings indicate that a relatively small but significant percentage of patients could benefit from this approach, especially when NGS tests are combined with IHC. Although the implementation of molecular tumor boards is still pending in Chile, this should be encouraged and incorporated into clinical practice.

## Figures and Tables

**Figure 1 jpm-12-00195-f001:**
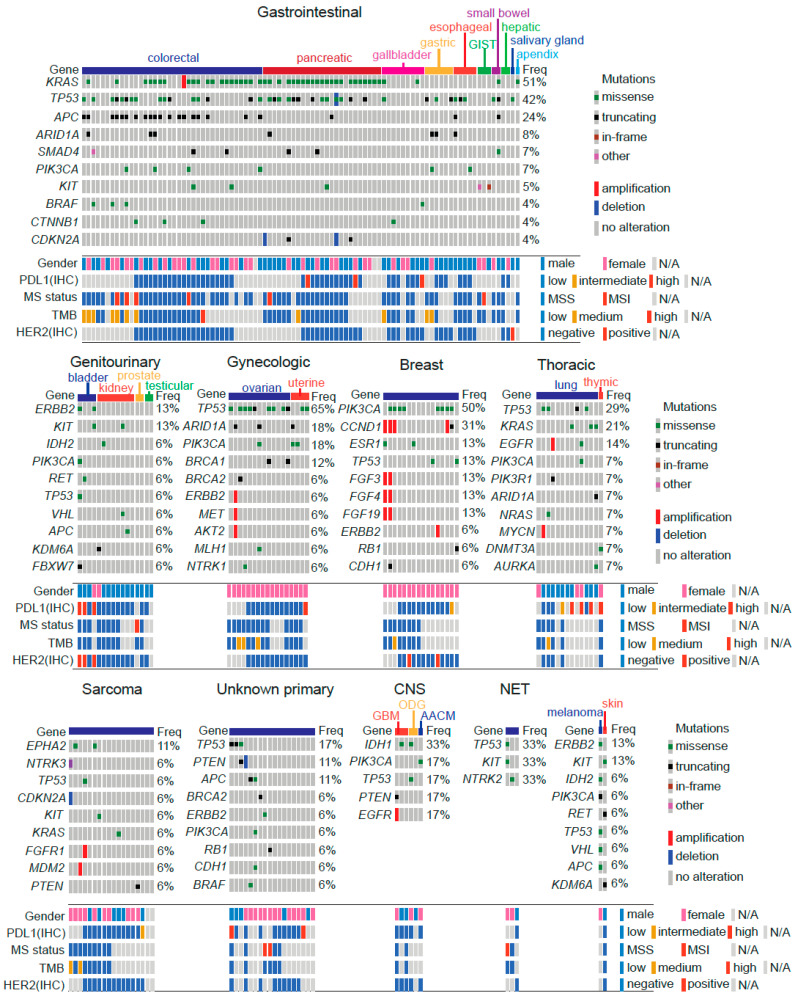
Mutational landscape of 202 Chilean advanced stage refractory cancer patients. Tumors were grouped as: gastrointestinal, genitourinary, gynecologic, breast, thoracic, sarcoma, unknown primary, central nervous system (CNS), neuroendocrine tumors (NET), melanoma and skin. Abbreviations: GIST: gastrointestinal stromal tumor; MS: microsatellite; TMB: tumor mutational burden; GBM: glioblastoma multiforme; ODG: oligodendroglioma; AACM: anaplastic astrocytoma.

**Figure 2 jpm-12-00195-f002:**
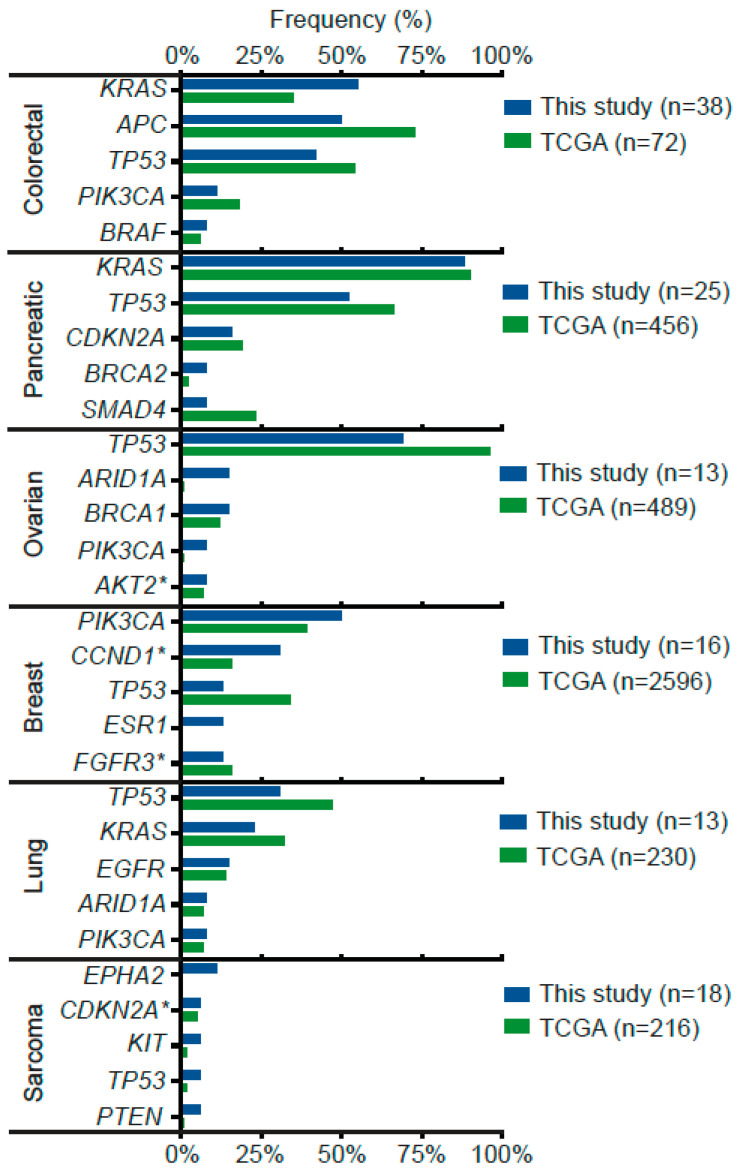
Most frequent gene alterations (top 5) by cancer type in our cohort versus TCGA cohorts. * Indicates copy number alteration, amplification or deletion.

**Figure 3 jpm-12-00195-f003:**
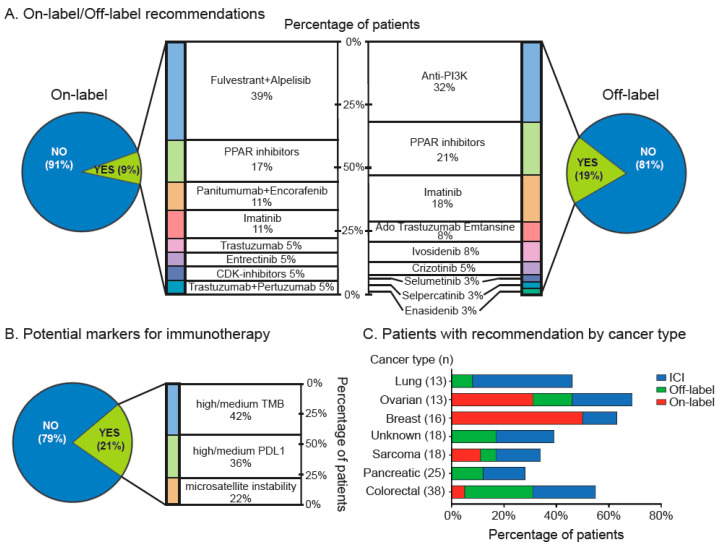
Clinical actionability for targeted therapies. (**A**) On-label (left panel) and off-label (right panel) and (**B**) immune checkpoint inhibitor recommendations based in NGS/IHC profiles. (**C**) Percentages of recommendation (on-label, off-label or ICI) by cancer type.

**Table 1 jpm-12-00195-t001:** Distribution of cancer types in this cohort.

Cancer Type	*n* (%)
Colorectal	38 (18.8)
Pancreatic	25 (12.4)
Sarcoma	18 (8.9)
Unknown	18 (8.9)
Breast	16 (7.9)
Ovarian	13 (6.4)
Lung	13 (6.4)
Gallbladder/bile duct	9 (4.5)
Kidney	8 (4)
Gastric	6 (3)
CNS *	6 (3)
Esophageal	5 (2.5)
Uterine	4 (2)
Bladder	4 (2)
Neuroendocrine	3 (1.5)
GIST *	3 (1.5)
Prostate	2 (1)
Small bowel	2 (1)
Hepatic	2 (1)
Testicular	2 (1)
Melanoma	1 (0.5)
Appendix	1 (0.5)
Thymic	1 (0.5)
Non-melanoma/skin	1 (0.5)
Salivary gland	1 (0.5)
Total	202 (100)

* Abbreviations: CNS: central nervous system; GIST: gastrointestinal stromal tumors.

## Data Availability

The data presented in this study are available on request from the corresponding author.

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
