# Peer review of "Mutational Landscape and Actionable Target Rates on Advanced Stage Refractory Cancer Patients: A Multicenter Chilean Experience"

_jpm, 2022, doi:10.3390/jpm12020195_

Round 1

Reviewer 1 Report

Sample is very heterogeneous. For most types of tumors less than 5 cases were studied. For frequent types of cancers (colon, breast, lung) the number is too low to generate new information.

Were any of the samples subjected to single gene testing before NGS, for example lung cancer (EGFR, ROS1, ALK, etc.), colorectal cancer (RAS testing).

The author should provide information regarding the type of tumors in which additional studies analysis were performed and criteria to perform the test (HER2, MSI, etc.).

The author should provide more information regarding the different panels used Where all they based on amplicons or on hybrid capture? Were all able to detect CNVs and fusions?. Probably not. So the frequencies of amplification or fusions provided are lower than expected and comparison with TCGA is not appropriate. This reviewer has obtained little information in the provider’s web page about the most frequent panels used (OncoDeep). Is it an IVD validated test? The authors should provide information regarding the type of tumor studied for the different panels. Some test seemed to be used on liquid biopsy, but the authors do no specify this point.

The authors did not discuss differences between their finding and TCGA. In addition to difference due to panel characteristics, as previously mentioned, stage and previous treatments are probably other explanations. For example, the higher frequency of ESR1 mutations in this series of breast cancers is probably due to the inclusion of ER+ tumors treated with hormone therapy.

The author suggested that one of the strengths of this study is its regional/ethnic nature. The authors claimed “our report is the first of its kind within the region delivering real-world evidence on the mutational landscape in a cohort 216 of 202 Latin American-ancestry advanced stage cancer patients”. From an ethnic point of view, Chilean population is very heterogeneous with European ancestry predominance. In addition, this study was conducted by private institutions that probably predominantly serve people of high socioeconomic level, where European ancestry is even higher. Unfortunately, the study does not provide specific ethnic information of the patients.

Author Response

We would like to thank you for taking the time to review our work. Below we have prepared point by point answers to all your comments and requests. We trust this new revised version of our manuscript is suitable for publication

Reviewer #1

1-Sample is very heterogeneous. For most types of tumors less than 5 cases were studied. For frequent types of cancers (colon, breast, lung) the number is too low to generate new information.

R1- Correct. We agree with the reviewer. We acknowledge that our work has several limitations, these are listed in the discussion section and include: heterogeneity of samples and low numbers for highly prevalent cancers among others (starting at LINE 288). Our intention was to simply collect real world evidence in this regard and to deliver an estimate of patient that could benefit from targeted therapies

2-Were any of the samples subjected to single gene testing before NGS, for example lung cancer (EGFR, ROS1, ALK, etc.), colorectal cancer (RAS testing).

R2- Interesting question. We are adding two new supplementary tables (new Supplementary Tables S3 and S4) with a small subset of patients (n=35) and more information including status of other markers: MSI for pancreatic and colorectal cancers; KRAS, NRAS and BRAF for colorectal; KIT and PDGFRA status for GISTs; EGFR, ALK and PDL1 for lung cancers; and HER2 for gastric cancers (Supplementary Table S4). Also, we report separately 4 patients treated with targeted therapy based in molecular testing (Supplementary Table S3)  

3-The author should provide information regarding the type of tumors in which additional studies analysis were performed and criteria to perform the test (HER2, MSI, etc.).

R3- See reply R2 above

4-The author should provide more information regarding the different panels used Where all they based on amplicons or on hybrid capture? Were all able to detect CNVs and fusions?. Probably not. So the frequencies of amplification or fusions provided are lower than expected and comparison with TCGA is not appropriate. This reviewer has obtained little information in the provider’s web page about the most frequent panels used (OncoDeep). Is it an IVD validated test?

R4- The reviewer raises a couple of very interesting points here. First, as the reviewer points out there is a heterogeneity of tests used in our work. As requested, we are including a new Supplementary Table (S1) that summarizes main technical features for each test, including amplicons/hybridization based technologies. Secondly, we acknowledge that not all used platforms in our study included gene fusions. However, they all included Copy Number Alterations (CNAs). Please note that in our comparisons with TCGA data (Fig. 2) we did not use genes that displayed fusions; therefore, we believe this approach is valid and sound. Third, regarding the OncoDEEP test, the information provided by the manufacturer at the company’s website (https://www.oncodna.com/en/healthcare-providers/biomarker-tests/oncodeep/) indicates the following information:

“Is OncoDEEP® accredited and certified?

The ISO 15189 (Medical laboratories-Requirements for quality and competence), CE-IVD (In vitro diagnostic devices complied to be sold in Europe), ISO 27001 (Information security management) and ISO 13485:2016 (Quality Management System) apply to our OncoDEEP® comprehensive biomarker test.” OncoDEEP is CE-IVD certified

5-The authors should provide information regarding the type of tumor studied for the different panels. Some test seemed to be used on liquid biopsy, but the authors do no specify this point.

R5- As requested, we are including a new supplementary table (Supplementary Table S2) along with the revised manuscript that indicates cancer type, NGS panel utilized and sample type for each patient (tumor type). In summary, 184 were tumor samples, 15 were liquid biopsy (ctDNA) and 3 were both (liquid and tumor sample)

6-The authors did not discuss differences between their finding and TCGA. In addition to difference due to panel characteristics, as previously mentioned, stage and previous treatments are probably other explanations. For example, the higher frequency of ESR1 mutations in this series of breast cancers is probably due to the inclusion of ER+ tumors treated with hormone therapy.

R6- OK. As requested by the reviewer we have added a new paragraph in the discussion section of the revised manuscript that incorporates these comments (starting at LINE 241)

7-The author suggested that one of the strengths of this study is its regional/ethnic nature. The authors claimed “our report is the first of its kind within the region delivering real-world evidence on the mutational landscape in a cohort 216 of 202 Latin American-ancestry advanced stage cancer patients”. From an ethnic point of view, Chilean population is very heterogeneous with European ancestry predominance. In addition, this study was conducted by private institutions that probably predominantly serve people of high socioeconomic level, where European ancestry is even higher. Unfortunately, the study does not provide specific ethnic information of the patients.

R7- This is another very interesting point. Unfortunately, this information was unavailable. We are incorporating the comments made by the reviewer as limitations of our study and specify that our results should be interpreted cautiously given the potential bias (starting at LINE 288)

Reviewer 2 Report

Dear Editor in Chief

The manuscript entitled “Mutational landscape and actionable target rates on advanced stage refractory cancer patients: A multicenter Chilean experience” is describing the usefulness of the mutational analysis of the refractory cancer patients in Chile. The manuscript is well written and the results are well presented. The discussion section is informative.

I found the manuscript is interesting despite the low number of cases. It describes the heterogeneity of cancers and the need for more investigation using new technologies such as NGS to reach the appropriate treatment options. However, I have some minor concerns about this work:

  1. Some types of cancer that are included in the study are not representative of that specific type of cancer due to the low number of cases.
  2. Most of the cases are colorectal and pancreatic which may affect the bias of some findings. For instance, breast cancer samples are very low compared to the prevalence of this cancer. It is obvious in Fig. How the percentage of TP53 and CCND1 when the results are compared with the TCGA. So the conclusion is better to specify this. 

I recommend the acceptance of this manuscript.

Author Response

We would like to thank you for taking the time to review our work. Below we have prepared point by point answers to all your comments and requests. We trust this new revised version of our manuscript is suitable for publication

Reviewer #2

The manuscript entitled “Mutational landscape and actionable target rates on advanced stage refractory cancer patients: A multicenter Chilean experience” is describing the usefulness of the mutational analysis of the refractory cancer patients in Chile. The manuscript is well written and the results are well presented. The discussion section is informative.

1- I found the manuscript is interesting despite the low number of cases. It describes the heterogeneity of cancers and the need for more investigation using new technologies such as NGS to reach the appropriate treatment options. However, I have some minor concerns about this work:

  1. Some types of cancer that are included in the study are not representative of that specific type of cancer due to the low number of cases.

R1- This is correct. We acknowledge the number of cases for many cancer types in our cohort are too small to be representative (listed as a limitation of our study). Our intention was to obtain real world data in this regard and to obtain an overall percentage of patients that could benefit from NGS and molecular tumor boards (starting at LINE 288)

  1. Most of the cases are colorectal and pancreatic which may affect the bias of some findings. For instance, breast cancer samples are very low compared to the prevalence of this cancer. It is obvious in Fig. How the percentage of TP53 and CCND1 when the results are compared with the TCGA. So the conclusion is better to specify this. 

R2- Agreed. We have incorporated these comments along with other limitations of our study in the discussion section of the revised manuscript

3- I recommend the acceptance of this manuscript.

R3- Thank you for your positive comments and for taking the time to review our work

Round 2

Reviewer 1 Report

The authors have answer some of the questions raised.